# Galaxy-ML: An accessible, reproducible, and scalable machine learning toolkit for biomedicine

**Qiang Gu[1,2], Anup Kumar[3], Simon Bray[3], Allison Creason[1,2], Alireza Khanteymoori[3], Vahid Jalili[1,2], Björn Grüning[3], Jeremy Goecks[1,2]***

**1** Department of Biomedical Engineering, Oregon Health & Science University, Portland, Oregon, United States of America, **2** The Knight Cancer Institute, Oregon Health & Science University, Portland, Oregon, United States of America, **3** Bioinformatics Group, Department of Computer Science, University of Freiburg, Freiburg, Germany

* goecksj@ohsu.edu

**Data Availability Statement:** Data are in publicly available analysis histories on the public European Union Galaxy server at https://usegalaxy.eu. Links to individual datasets and analyses are in S1 Table

## Abstract

Supervised machine learning is an essential but difficult to use approach in biomedical data analysis. The Galaxy-ML toolkit (https://galaxyproject.org/community/machine-learning/) makes supervised machine learning more accessible to biomedical scientists by enabling them to perform end-to-end reproducible machine learning analyses at large scale using only a web browser. Galaxy-ML extends Galaxy (https://galaxyproject.org), a biomedical computational workbench used by tens of thousands of scientists across the world, with a suite of tools for all aspects of supervised machine learning.

This is a *PLOS Computational Biology* Software paper.

## Introduction

Machine learning (ML) has become an essential tool in biomedicine to make sense of large, high-dimensional datasets such as those found in genomics, proteomics, and imaging [1–3]. In supervised machine learning, these datasets are used to build statistical models from high-dimensional feature sets that can predict continuous values (regression analysis) or discrete classes (classification). Example applications of ML to biomedicine include developing predictive models for drug metabolism rates using brain images [4,5], genotype-phenotype associations [3], and drug response in model systems [6,7]. Deep learning, which leverages multi-layer neural networks, has been used for prediction of splice sites [8], protein structures [9], and cancer diagnosis from histopathology images [10].

Despite these successes, machine learning is often difficult to use in biomedicine. A successful ML application to biomedical data spans from biological analysis tools to machine learning tools for feature engineering, model building, and evaluation. Integrating ML and biological analysis tools is critical because the biological tools are used to create the features, such as genomic variants and protein abundance levels, that are used in the predictive model. In addition to tool integration challenges, ML tools must also be easily accessible, scale to large

for use case 1, S3 Text for use case 2, and S4 Text for use case 3.

**Funding:** This work was supported by NIH grants HG006620 (JG), CA233280 (JG), and CA231877 (JG) and NSF Grant 1661497 (JG), with additional support from German Federal Ministry of Education and Research grant 031L0101C (BG). The funders had no role in study design, data collection and analysis, decision to publish, or preparation of the manuscript.

datasets, and reproducible. As the size and number of biomedical datasets continue to grow, computational infrastructure such as workflow engines, software package managers, and job schedulers are needed for scaling and reproducing machine learning applications in biomedicine. Addressing these challenges requires an integrated software solution that (1) makes machine learning accessible to biomedical scientists who have limited programming and informatics knowledge and (2) connects machine learning with the broader ecosystems of biomedical analysis tools and a scalable computational workbench.

To meet this need, we have developed Galaxy-ML (Fig 1), a toolkit for the Galaxy platform (http://galaxyproject.org) [11] that features a large and diverse suite of supervised machine learning tools. Galaxy is a user-friendly web-based computational workbench used by tens of thousands of scientists across the world for a wide variety of biomedical data analysis, including genomics, proteomics, metabolomics, cheminformatics, image processing, and flow cytometry. The goal of Galaxy-ML is to provide the worldwide Galaxy user community with the ability to incorporate machine learning into their analyses. Galaxy-ML has already gained substantial usage in the Galaxy community. Based on download statistics from the Galaxy ToolShed [12], a tool repository for Galaxy, we estimate that Galaxy-ML has been installed on 80 Galaxy servers across the world. Galaxy-ML tools have also been run more than 12,000

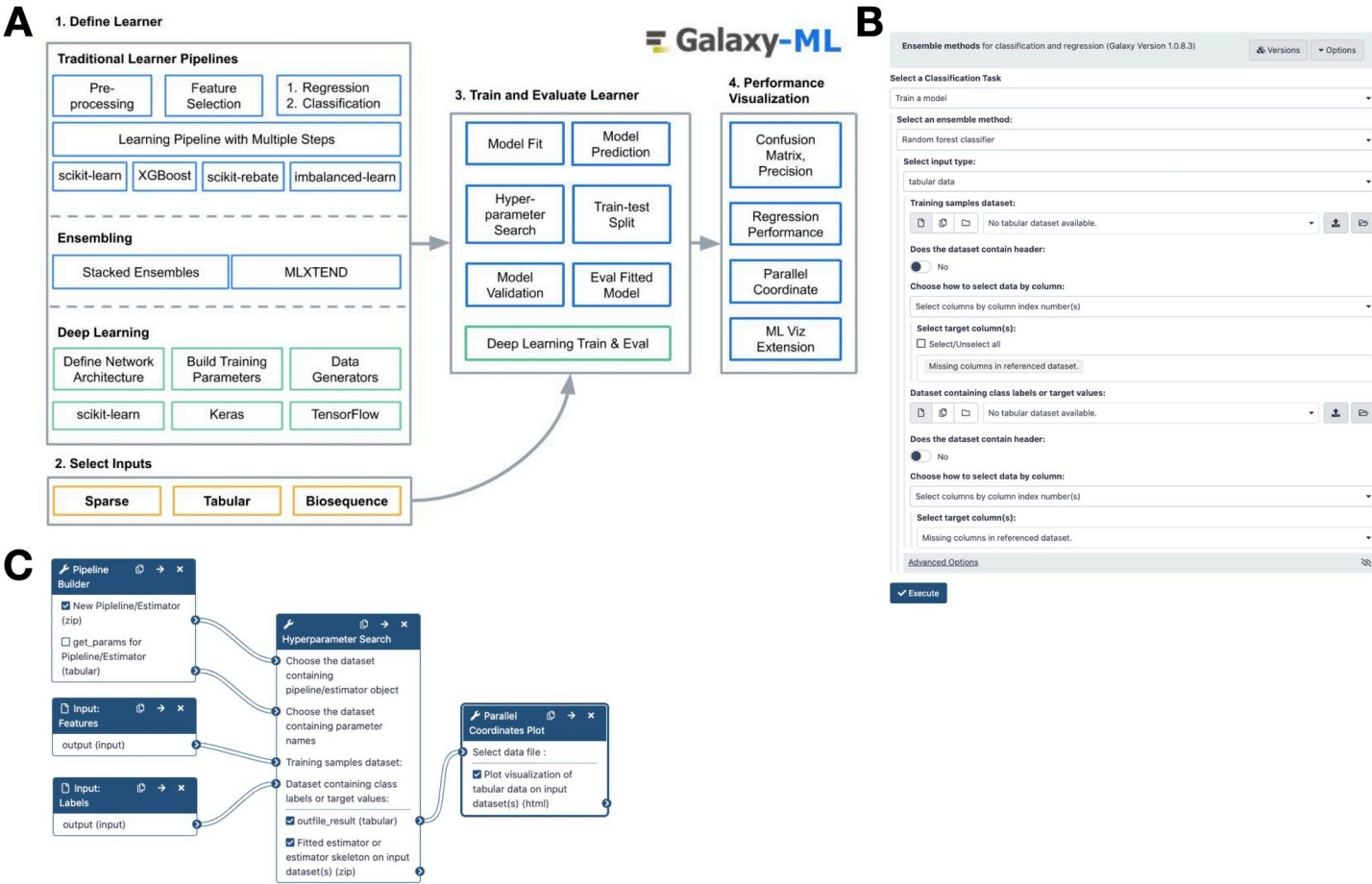

**Fig 1.** Panel A: The Galaxy-ML platform provides all the tools necessary to define a learner, train it, evaluate it, and visualize its performance. Panel B is a screenshot of the Galaxy tool to create a gradient boosted classifier. Panel C shows a Galaxy workflow to create a learner using a pipeline, perform hyperparameter search, and visualize the results.

times on the public U.S. server (https://usegalaxy.org, see "Machine Learning" section in the tool panel on the left or use tool search and type in "Machine Learning") and European Union Galaxy server (https://ml.usegalaxy.eu for the machine learning portal or https://usegalaxy.eu for the general portal with all tools).

A key aspect of Galaxy is its web-based user interface, enabling anyone to use complex analysis tools and multi-tool workflows without requiring detailed knowledge of workflows, software dependencies, or job schedulers. Galaxy-ML uses the Galaxy web interface to make machine learning tools and pipelines widely accessible. Iterative development is very common in machine learning, from engineering and selecting features to tuning model hyperparameters. With Galaxy's web interface and Galaxy-ML tools, it is simple to repeatedly perform some or all facets of machine learning, from feature engineering to model development and evaluation. Importantly, Galaxy-ML does not restrict what users can do: nearly all tools in Galaxy-ML are fully featured and provide the same level of flexibility that is found in their corresponding programmatic tools. Galaxy-ML tools and workflows can also be run programmatically via Galaxy's application programming interface (API), which may be preferred for large or automated analyses.

## Design and implementation

Galaxy-ML provides key benefits in scalability, reproducibility, and workflow development. Large machine learning analyses, such as optimizing hyperparameters and model evaluation across many different datasets, can require building tens of thousands of models. Galaxy-ML uses Galaxy's workflow system to execute large-scale analyses by distributing them across one or more computing clusters and running them in parallel. Galaxy ensures reproducibility by recording all parameters and tools used, so all analyses, including those for machine learning, are completely reproducible. This is critical, as reproducibility has become critical in machine learning research [13,14]. Galaxy-ML enables end-to-end machine learning analyses that begin with processing primary biological data and ends with trained machine learning models that can make predictions of phenotypic attributes like demographics or prognosis. For instance, this interactive tutorial [15] uses Galaxy-ML to reproduce a study that predicts an individual's chronological age from RNA-seq data [16,17]. End-to-end workflows are possible because Galaxy-ML's machine learning tools can be connected to any of the more than 7,800 tools available in the Galaxy ToolShed [12] for analyzing genomics, proteomics, imaging, and other kinds of biomedical data.

Galaxy-ML supports four major steps in machine learning—preprocessing, modeling, ensembling, and evaluation—by integrating six machine learning libraries (Table 1) together with additional visualization and conversion tools. Scikit-learn [18] provides the foundation for Galaxy-ML with approaches for all four major steps. Additional libraries are included to

**Table 1. Software libraries integrated into Galaxy-ML and their applications.**

| Software Library | Applications |
|---|---|
| Scikit-learn [18] | Various approaches for preprocessing, modeling, ensembling, and evaluation |
| Scikit-rebate [19] | Feature selection |
| Imbalanced-learn [20] | Approaches for working with imbalanced datasets |
| XGBoost [21] | Modeling using high-performance gradient boosting with decision trees or linear models |
| Keras [22] | Modeling using deep learning |
| Mlxtend [23] | Modeling using meta-ensembles |
| LightGBM [28] | Modeling using gradient boosting with the LightGBM algorithm |

meet key needs for machine learning in biomedicine, including feature selection, approaches for working with imbalanced datasets, and modeling approaches using gradient boosted decision trees, deep learning, and ensembling. Documentation, along with tutorials, is available at https://galaxyproject.org/community/machine-learning/, and links to the Galaxy-ML code and tool repositories are available in the Methods section.

## Results

We demonstrate the utility of Galaxy-ML in three use cases: (1) extending a machine learning benchmark experiment where 4,000 models were created and evaluated on 276 biomedical datasets [24]; (2) predicting drug response activity in cancer cell lines using gene expression datasets using stacked meta-ensembles; and (3) validating deep learning models for genomics that predict, among other attributes, the functional impact of genetic variants. The Methods section provides links to complete analysis histories and results so that all analyses can be fully reproduced on any Galaxy server with the Galaxy-ML tool suite. All analyses were performed on a public Galaxy server at https://usegalaxy.eu and are listed at https://ml.usegalaxy.eu. All workflows, data and results can be accessed via a web browser and analyses can be reproduced directly.

### Automatically creating and evaluating thousands of machine learning models

In the first use case, we used Galaxy-ML to extend an analysis of machine learning models across 276 biomedical datasets [25]—164 classification datasets and 112 regression datasets [25]. The original analysis compared performance of 13 models on the 164 classification datasets. We applied 15 models to the classification datasets and 14 models to the regression datasets, creating a total of 4,028 trained models with hyperparameters optimized using grid search (S1 Text and S1 Table). We evaluated all models using 10-fold cross-validation (CV). Because many datasets were imbalanced, *F1* scoring rather than ROC AUC was used to evaluate performance of classification models, and Pearson's $R^2$ was used to evaluate performance of regression models. Performance of classification models are concordant with the initial publication: (i) boosted tree models perform best overall (Fig 2A) and (ii) automated hyperparameter optimization improves performance for many models (Fig 2B). Performance of regression models are similar to those in classification, though boosted tree models only modestly outperform random tree models, and hyperparameter optimization often improves results most for models with low overall performance (S2 Text and S1 Fig).

### Developing meta-ensembles for predicting cell line drug response

For the second use case, we used Galaxy-ML to apply sophisticated machine learning models, including stacked meta-ensemble predictors, to predict drug response in cancer cell lines from high-throughput gene expression data from RNA-seq (S3 Text). Because cancer cell lines serve as models for patient tumors, accurate predictions of drug response can be used to improve understanding of cancer systems biology and inform patient treatment recommendations [26]. Gene expression and drug response data was obtained from DepMap [27]. There are two key challenges for this dataset: (1) there are ~50,000 gene expression features but only ~1,000 cancer cell lines and ~700 drugs, so preventing overfitting is essential, and (2) the dataset is highly imbalanced because there is a small number of cell lines that respond to each drug.

Using Galaxy-ML, we built a meta-ensemble as well as other learners for each drug. The meta-ensemble included a linear boosted model, tree boosted model, and k-nearest neighbor regression, and principal component analysis (PCA) was used for dimensionality reduction in several learners. Dimensionality reduction was used to address the challenge of using a dataset

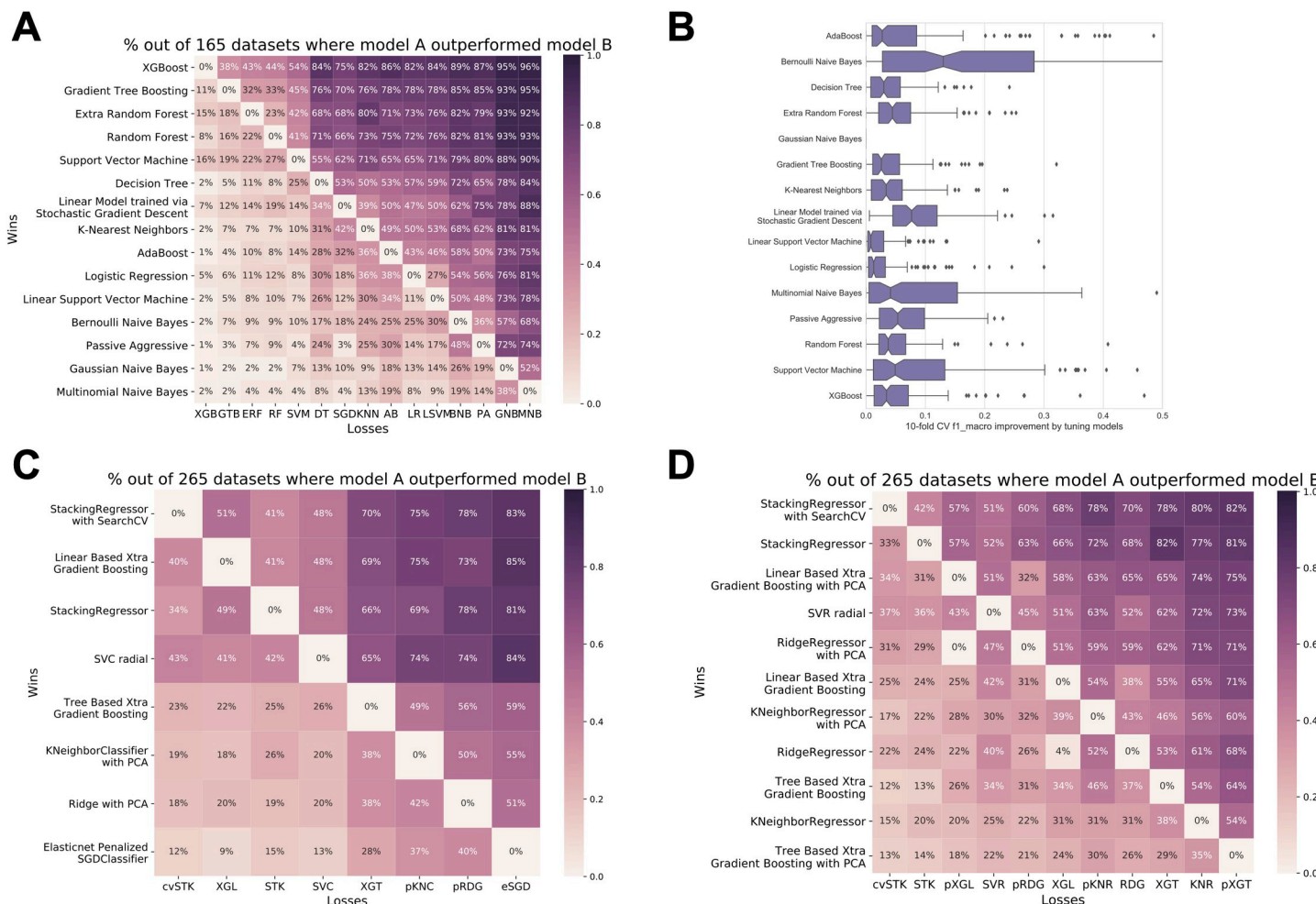

**Fig 2. Pairwise performance comparisons for use cases 1 and 2.** Use case 1 pairwise comparisons for classification tasks on 164 structured biomedical datasets [25] show decision tree forests perform best (panel A) and hyperparameter optimization can improve the performance of most models (panel B). Use case 2 results for prediction using regression (panel C) and classification (panel D) show ensemble approaches that use stacking perform best, though linear-based gradient boosting also performs. In panels A, C, and D, heatmaps show the percentage of datasets for which the model listed along the row outperforms the model along the column. For instance, in panel A, XGBoost outperforms Gradient Tree Boosting (GTB) from scikit-learn on 38% of datasets, GTB outperforms XGBoost on 11% of datasets, and they perform equivalently on 51% of datasets.

with a very large number of features. We developed predictors for both regression and classification; labels for classification were generated by thresholding drug response values and labeling cell lines as responders or non-responders to each drug using a cutoff of z-score < -1 for responders. Predictors were scored using average precision to address the challenge of assessing model performance on a highly imbalanced dataset, where the goal is to identify responders (true positives) amongst a very large number of non-responders. To compare regressors and classifiers, average precision for regressors was calculated using rank-ordered predictions, which has been done in past machine learning work in this space [6]. We evaluated each learner using nested CV, with 5-fold CV for 4 repetitions for the outer splits and 5-fold CV with two repetitions for the inner splits. Our results show that stacking regressors performed best for both regression (Fig 2C) and classification (Fig 2D). Linear boosting approaches also performed very well, with results that were on par with the meta-ensembles. Successful completion of these two use cases shows that Galaxy-ML can support large and diverse machine learning experiments.

### Reproducing deep learning models for DNA sequence analysis

In the third use case, Galaxy-ML was used to validate key results from Selene [28], a deep learning toolkit for biological sequence data built on the PyTorch library. Using Galaxy-ML, we reimplemented two deep learning architectures originally implemented in Selene that model and predict regulatory elements, including transcription factor binding sites, DNase I hypersensitive sites, and histone marks (S4 Text). Results from these models are within 1% of those reported for Selene (Figs 3 and S2 and S3 and S2 Table). Critical to this work was the implementation of data generators in Galaxy. Data generators meet two important needs: (1) producing new examples from existing data to increase the number of instances available for training, and (2) feeding small sets of examples to the deep learning model so that the entire training dataset does not need to be loaded into memory. This use case demonstrates that Galaxy-ML deep learning tools are general and powerful enough to support realistic use cases and that Selene results validated across different deep learning implementations.

### Reproducibility and extensibility

All analyses in Galaxy are highly reproducible from individual tool executions to complete workflows because Galaxy records and stores all parameter settings, tool versions, and

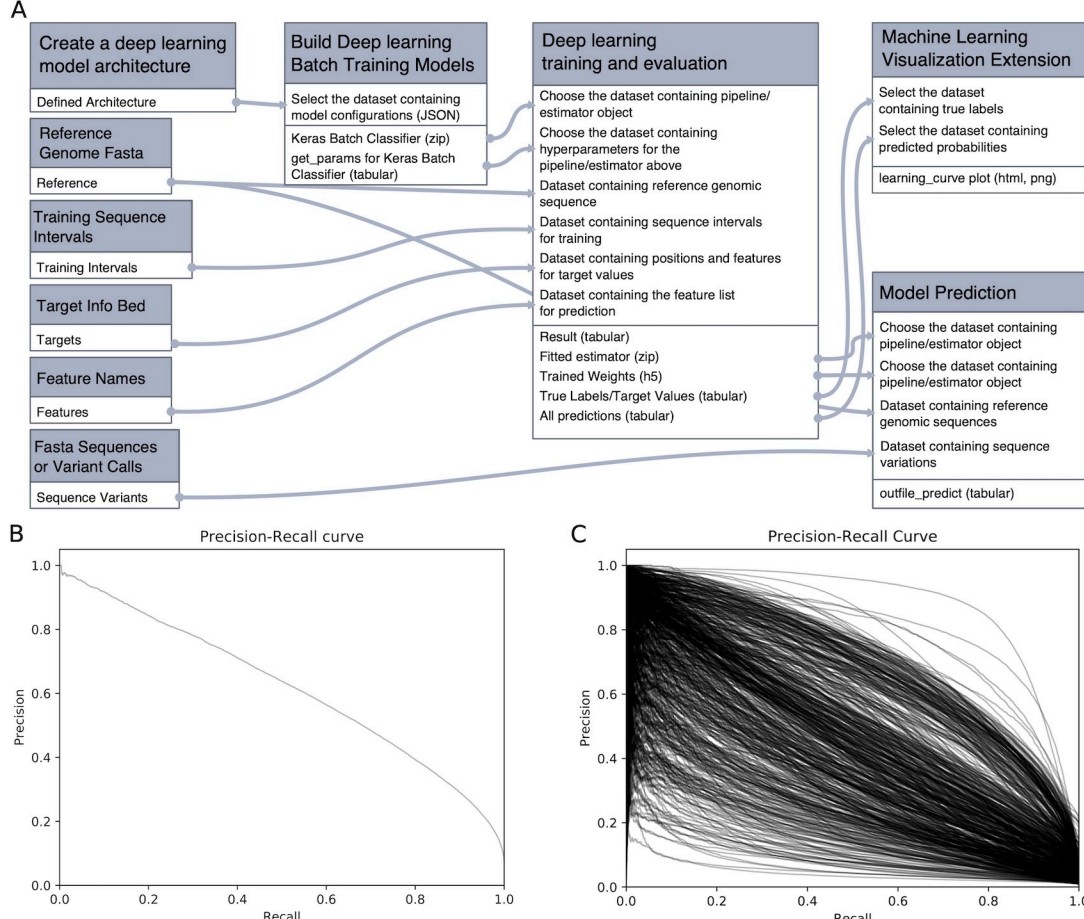

**Fig 3.** (A) Galaxy workflow to create and train a deep learning model, then use the model for visualization and prediction. (B) Precision-recall curve for a deep neural network trained to predict binding sites for a single transcription factor. (C) Precision-recall curves for a deep neural network that predicts 919 regulatory element binding profiles, with each curve in the plot denoting a precision-recall curve for 1 regulatory element.

workflow versions used in an analysis. This reproducibility extends to all Galaxy-ML tools as well as Galaxy workflows that include Galaxy-ML tools. There are ongoing efforts to enable reproducibility of Galaxy workflows outside of the Galaxy platform via interoperable workflows. The primary focus of these efforts are to make Galaxy compatible with workflows written in Common Workflow Language (CWL) [29]. CWL is a community standard for workflow definitions that is embraced by many workflow engines. When Galaxy's CWL features are complete, Galaxy will be able to execute CWL workflows as well as export Galaxy workflows in CWL format. Once in CWL format, Galaxy workflows can be executed in any workflow engine that supports CWL, and this interoperability will extend to workflows composed of Galaxy-ML tools.

It is possible to extend Galaxy-ML with additional machine learning software libraries and custom methods. Galaxy-ML has tools for data preprocessing, data generators, model definition, model training, and model evaluation, providing clear integration points where additional machine learning approaches can be added based on their functionality. These integration points mirror the `scikit-learn` application programming interface, which is widely used across the machine learning community. For instance, if a new library for creating gradient boosted decision trees becomes available, tools such as `Model Fit` and `Hyperparameter` Search can be augmented so users can create and use models from this library. As part of the use cases in the previous sections, we implemented custom modules for preprocessing, modeling, data splitting, and evaluation.

## Availability and future directions

The website https://galaxyproject.org/community/machine-learning/ provides a hub for machine learning in Galaxy and access to all Galaxy-ML tools, workflows and tutorials. We anticipate that this hub will serve as a community starting point to foster accessible machine learning in biomedicine. The Galaxy tool wrappers for our machine learning suite are available at the following URLs: (1) main tools: https://github.com/bgruening/galaxytools/tree/master/tools/sklearn and (2) utilities and custom classifiers: https://github.com/goeckslab/Galaxy-ML, and the entire suite can be installed onto any Galaxy server through the Galaxy ToolShed at http://bit.ly/galaxy-ml-toolshed.

Galaxy-ML accelerates biomedical research by making machine learning more accessible, scalable, and reproducible. We applied Galaxy-ML in three complex use cases that yielded novel insights from several large and diverse biomedical datasets. Galaxy-ML's tools are completely generalizable and have applications well beyond these use cases. With Galaxy's web-based user interface, an entire machine learning pipeline from normalization, feature selection, model definition, hyperparameter optimization, and cross-fold evaluation can be created and run on large datasets in parallel across a computing cluster using only a web browser. This makes scalable and reproducible machine learning accessible to biomedical scientists regardless of their informatics skills. By leveraging the more than 7,800 analysis tools available in Galaxy, comprehensive end-to-end analyses can be performed, which begins with primary analysis of -omics, imaging, or other large biomedical dataset and continues to downstream machine learning tools that build and evaluate predictive machine learning models from features extracted from the primary data.

Looking forward, additional machine learning tools, libraries, and datasets will be integrated into Galaxy-ML. High priority work includes support for deep learning with imaging datasets and integration of predefined and pretrained models. Future versions of Galaxy-ML will also include additional integration points to make it easier to implement and use new machine learning libraries and individual tools.

## Supporting information

**S1 Text. Use Case 1: PennML benchmark.**
(DOCX)

**S2 Text. Regression analysis: Comparison of 14 regressors on 112 Penn regression datasets.**
(DOCX)

**S3 Text. Use Case 2: DepMap Cancer Cell Lines.**
(DOCX)

**S4 Text. Use Case 3: Deep Learning for Genomics using Selene.**
(DOCX)

**S1 Table. A list of Galaxy histories and workflows used for the benchmarks in use case 1.**
Each history/workflow ensures that an analysis can be completely reproduced because it lists all analysis steps and parameters. Each algorithm runs with two parameter configurations: default and best. Default configuration is a default value of parameters in Galaxy toolbox and best parameters are obtained by hyperparameter optimization.
(DOCX)

**S2 Table. Performance results obtained using Galaxy-ML models fully trained using GPU and Selene models.** All datasets used were obtained from Selene. AUPRC is the area under the precision-recall curve, and is also known as the average precision. "N.R." means that the models did not report this information.
(DOCX)

**S1 Fig. Comparison of different regression models.** In panels A and C, heatmaps show the percentage of datasets for which the model listed along the row outperforms the model along the column. Panel A shows a heatmap in which each square contains a number of datasets for which the regressor on the left (wins) performed better than the regressor on the bottom (losses). For example, by mapping the color of the square between adaboost (shown on y-axis) and linear regression (LR) shown on x-axis to the adjacent color-scale, we conclude that the adaboost regressor performs better on 75–80 datasets (out of 112) than the linear regressor. The subplot also shows a comparison of different regressors (on y-axis). The ensemble regressors perform better on average than the other categories which include linear, tree and nearest neighbors regressors. Panel B shows a comparison between the running time and accuracy of different regressors. We compute an average running time of each regressor over all 112 datasets. The running time of a regressor on a dataset is the sum of the training and validation times for the best regression model. The regressors such as xgboost, gradient boosting and extra trees achieve > 0.80 r-squared score, but extra trees regressor requires significantly more time to finish compared to the other two regressors. Regressors such as linear regression, huber and elastic net are fast, but their accuracy is low. Decision and extra tree regressors are also fast, but their accuracy is better (> 0.7 r-squared score) than the linear regressors. Panel C shows the r-squared scores of each regressor for all datasets. The linear regressors at the bottom-left of the subplot achieve lower scores than the ensemble regressors such as xgboost, gradient boosting at top-left of the subplot. We can also see that for a few datasets, none of the regressors perform well. Panel D shows the importance of tuning the hyperparameters of the regressors for each dataset. It is not recommended to compute the performance of predictive algorithms over multiple datasets using the same or default values of their hyperparameters. The performance of a regressor varies for different values of hyperparameter for a dataset. Therefore, we computed the best set of values of hyperparameters for each dataset using an

exhaustive search strategy (grid-search). The figure shows an improvement in r-squared scores for each regressor due to hyperparameter optimisation. Regressors such as elastic net, k nearest neighbours, decision tree and linear svr show higher improvements than bagging, random forest, adaboost, gradient boosting, xgboost, extra trees, linear regression, huber and gradient boosting in their respective r-squared scores averaged over all 112 datasets.
(TIFF)

**S2 Fig. Visualized results obtained using the DeepSEA architecture to model regulatory elements for a single tissue-specific transcription factor.**
(TIFF)

**S3 Fig.** Visualized ROC curve results obtained from using the extended DeepSEA architecture [30] to model 919 regulatory elements: (A) average ROC for all elements and (B) individual ROC curves for each element.
(TIFF)

## Author Contributions

**Conceptualization:** Qiang Gu, Anup Kumar, Simon Bray, Björn Grüning, Jeremy Goecks.

**Data curation:** Qiang Gu, Anup Kumar, Simon Bray, Allison Creason, Alireza Khanteymoori, Vahid Jalili.

**Formal analysis:** Qiang Gu, Anup Kumar, Simon Bray, Allison Creason, Alireza Khanteymoori, Vahid Jalili.

**Funding acquisition:** Björn Grüning, Jeremy Goecks.

**Investigation:** Qiang Gu, Anup Kumar, Simon Bray, Allison Creason, Alireza Khanteymoori, Björn Grüning.

**Methodology:** Qiang Gu, Anup Kumar, Simon Bray, Allison Creason, Alireza Khanteymoori, Vahid Jalili, Björn Grüning, Jeremy Goecks.

**Project administration:** Björn Grüning, Jeremy Goecks.

**Resources:** Qiang Gu, Anup Kumar, Allison Creason, Björn Grüning, Jeremy Goecks.

**Software:** Qiang Gu, Anup Kumar, Simon Bray, Allison Creason, Alireza Khanteymoori, Björn Grüning.

**Supervision:** Björn Grüning, Jeremy Goecks.

**Validation:** Qiang Gu, Anup Kumar, Simon Bray, Vahid Jalili, Björn Grüning, Jeremy Goecks.

**Visualization:** Qiang Gu, Anup Kumar, Simon Bray, Allison Creason, Alireza Khanteymoori, Vahid Jalili, Björn Grüning, Jeremy Goecks.

**Writing – original draft:** Qiang Gu, Anup Kumar, Simon Bray, Allison Creason, Alireza Khanteymoori, Vahid Jalili, Björn Grüning, Jeremy Goecks.

**Writing – review & editing:** Qiang Gu, Jeremy Goecks.

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
