## [Decision Letter · Decision Letter 0]

21 Mar 2021

Dear Dr. Goecks,

Thank you very much for submitting your manuscript "Galaxy-ML: An Accessible, Reproducible, and Scalable Machine Learning Toolkit for Biomedicine" for consideration at PLOS Computational Biology. As with all papers reviewed by the journal, your manuscript was reviewed by members of the editorial board and by several independent reviewers. The reviewers appreciated the attention to an important topic. Based on the reviews, we are likely to accept this manuscript for publication, providing that you modify the manuscript according to the review recommendations.

Sincerely,

Mihaela Pertea

Software Editor

PLOS Computational Biology

Mihaela Pertea

Software Editor

PLOS Computational Biology

[LINK]

Reviewer's Responses to Questions

**Comments to the Authors:**

Reviewer #1: Overview: Well written paper. Clear and concise. The Galaxy-ML toolkit and it's accessibility through Galaxy will be a powerful tool available to Biomedical scientists. The paper clearly demonstrates multiple examples of how the toolkit can be used by reproducing existing ML data models with accuracy.

Major Comments: None

Minor Comments: The Galaxy platform itself has some limitations in terms of reproducibility. Although, the platform is openly available and hence the ML toolkit, it would be nice to see a more interoperable way of representing Galaxy workflows so they are repeatable in other cloud or local compute environments.

Extensibility is not necessarily clear or demonstrated in the publication. An potential minor addition, would be to address how additional ML libraries can be added to the existing toolkit and whether customization of those libraries is possible.

Conclusion: The paper is well written and the power of the toolkit as well as it's accessibility through Galaxy is successfully demonstrated as a novel addition to the growing resources available to the Biomedical community.

Reviewer #2: The authors describe Galaxy-ML, an extension to the Galaxy computational workbench that is designed to assist users with running supervised machine learning experiments in the cloud.

The Galaxy interface leverages popular machine learning frameworks including Scikit-learn and Keras to prepare, model, and evaluate supervised classifiers in the Galaxy setting.

The authors performed analyses and meta-analyses on example datasets, accurately describing rationale and methods for feature selection, as well as training and evaluating models.

The most valuable content in this paper comes from the methods section, which serves as an outstanding introductory guide for using the Galaxy-ML toolkit for exploratory machine learning analyses on diverse datasets. The provided workflows are an immense value add, as is the extensive community documentation cited in this manuscript at https://galaxyproject.org/community/machine-learning/.

Overall, this manuscript represents a valuable addition to the field of open-source and community-accessible tools for machine learning in the biomedical domain. The introduction to the tools and the provided methods and documentation are outstanding. I have only a couple concerns over specific statements made in the paper, which I would like to see the authors address before having this manuscript published in PLOS Computational Biology.

Concerns:

The statement “Galaxy-ML tools have also been run more than 12,000 times on the public U.S. (https://usegalaxy.org) and European Union Galaxy servers (https://usegalaxy.eu)” describes this toolkit as being run on both the US and EU servers, but I was unable to locate Galaxy-ML on the US servers with a moderate effort. The machine learning extensions for the EU server are located at https://ml.usegalaxy.eu, which I think is a useful URL the authors should consider adding to the manuscript, perhaps in lieu of the references to the primary server interfaces in this statement (which reference a broader set of tools for the EU server, and do not reference the Machine Learning tools at all on the US server). I would recommend the authors work with the US server team to incorporate the Galaxy-ML tools on that site as well, or provide some discussion in the manuscript on why they are not available on the US server. If the Galaxy-ML tools are not publicly available on the US server, the above statement should also be adjusted to reflect the reality of the situation (it is not useful to know that the authors somehow ran these tools on the US server if that is not available to the readers of this article).

The statement “To the best of our knowledge, this is among the most comprehensive machine learning experiments on cancer cell line RNA-seq and drug response datasets” is underqualified (to the best of our knowledge? among the most? how is comprehensiveness measured?), unnecessary (this does not add value to the paper or the useful work described there), and difficult to prove (large dataset ML experiments are a common research activity, particularly in cancer cell line data; many remain unpublished). This statement should be removed.

**Have all data underlying the figures and results presented in the manuscript been provided?**

Reviewer #1: None

Reviewer #2: Yes

PLOS authors have the option to publish the peer review history of their article (what does this mean?). If published, this will include your full peer review and any attached files.

Reviewer #1: No

Reviewer #2: **Yes: **Alex Handler Wagner

Figure Files:

Data Requirements:

Reproducibility:

References:

---

## [Editor Report · Decision Letter 1]

27 Apr 2021

Dear Dr. Goecks,

We are pleased to inform you that your manuscript 'Galaxy-ML: An Accessible, Reproducible, and Scalable Machine Learning Toolkit for Biomedicine' has been provisionally accepted for publication in PLOS Computational Biology.

Best regards,

Mihaela Pertea

Software Editor

PLOS Computational Biology

Mihaela Pertea

Software Editor

PLOS Computational Biology

---

## [Editor Report · Acceptance letter]

21 May 2021

PCOMPBIOL-D-21-00079R1 

Galaxy-ML: An Accessible, Reproducible, and Scalable Machine Learning Toolkit for Biomedicine

Dear Dr Goecks,

I am pleased to inform you that your manuscript has been formally accepted for publication in PLOS Computational Biology. Your manuscript is now with our production department and you will be notified of the publication date in due course.

With kind regards,

Katalin Szabo
